# Assessing the effectiveness and implementation of a universal classroom-based set of educator practices to improve preschool children's social-emotional outcomes: Protocol for a cluster randomized controlled type 2 hybrid trial in Singapore

**Evelyn S. Tan**[1,2]*, **Bryce D. McLeod**[1,3], **Robyn A. Mildon**[1,2], **Aron Shlonsky**[4], **Cheryl K. F. Seah**[1,2], **Keri McCrickerd**[1,5], **Esther Goh**[1,2], **Gayatri Kembhavi**[1,2]

**1** Centre for Holistic Initiatives for Learning and Development, Yong Loo Lin School of Medicine, National University of Singapore, Singapore, Singapore, **2** Centre for Evidence and Implementation, Melbourne, Victoria, Australia, **3** Department of Psychology, Virginia Commonwealth University, Richmond, VA, United States of America, **4** Department of Social Work, Monash University, Clayton, Victoria, Australia, **5** Singapore Institute for Clinical Sciences (SICS), Agency for Science, Technology and Research (A*STAR), Singapore, Singapore

* evelyn.tan@partner.nus.edu.sg

## Abstract

### Background

Providing high-quality early childhood care and education is understood as key to maximizing children's potential to succeed later in life, as it stimulates young children's development of skills and competencies needed to promote optimal outcomes and success later in life. Despite the government's efforts to support the early childhood sector, educators in Singapore continue to report difficulties in implementing practices in classrooms that promote children's social, emotional, and cognitive development. To enhance educators' skills in these domains, we developed the Enhancing And Supporting Early development to better children's Lives (EASEL) Approach, a set of universal, educator-delivered practices for use with 3-6-year-old children in early childhood settings to improve social, emotional, behavioral, and executive functioning (SEB+EF) outcomes.

### Methods

This study will evaluate the effectiveness and implementation of the EASEL Approach in improving early childhood educators' teaching practices and, in turn, children's SEB+EF outcomes. We will conduct a cluster randomized controlled trial with a type 2 hybrid effectiveness-implementation study in 16 childcare centers. The EPIS (Explore, Prepare, Implement, Sustain) Framework will be used to inform the implementation of the EASEL Approach. Implementation strategies include training, educator self-assessments, practice-based coaching, and data monitoring. Our primary outcome is educators' teaching

**Data Availability Statement:** No datasets were generated or analysed during the current study. All relevant data from this study will be made available upon study completion.

**Funding:** This work was supported by an endowment from the Lien Foundation to the Yong Loo Lin School of Medicine, National University of Singapore.

**Competing interests:** The authors have declared that no competing interests exist.

practices. Secondary outcomes include educators' adoption of the EASEL Approach in everyday practice, the acceptability and feasibility of the EASEL Approach, and children's SEB+EF outcomes. Quantitative and qualitative data will be collected at baseline, six months, and after implementation.

## Conclusion

Findings from this study will provide significant evidence on the effectiveness of the EASEL Approach in improving educators' teaching practices and its impact on children's SEB+EF outcomes and the implementation of the EASEL Approach in early childhood classrooms in Singapore.

## Trial registration

This study was prospectively registered on ClinicalTrials.gov, Identifier: NCT05445947 on 6th July 2022.

## Introduction

The Lancet's series on advancing early childhood development [1] identified providing high-quality early childhood care and education as one of the main factors that can maximize children's potential to succeed in later life, particularly children from vulnerable or disadvantaged backgrounds. High-quality early childhood care and education are widely understood to be necessary for stimulating children's development of skills and competencies across academic and non-academic areas, which increases school readiness and has long-reaching impacts on outcomes later in life [2–7].

A holistic approach to early learning and development integrates children's social-emotional development, behavioral competency, and executive functioning. Social-emotional development and behavioral competency (SEB) refer to children's ability to understand, experience, express, and regulate their emotions and behaviors, and promote their ability to develop meaningful relationships with others [8]. The development of children's SEB significantly overlaps with executive functioning (EF), a set of cognitive processes necessary for regulating behavior [5]. Promoting the development of skills and competencies in these domains in the early years is essential for success in children's everyday interactions and subsequent development. In recognition of the importance of children's early development, there have been changes to the public policy agenda in Singapore to increase the quality of early childhood care and education [7, 9]. This has been supported by increased funding for the early childhood care and education sector and the establishment of a national regulatory agency for the early childhood sector, the Early Childhood Development Agency (ECDA).

Despite efforts by the government to support the early childhood sector in Singapore, several challenges remain. The Vital Voices for Vital Years 2 report [10] revealed difficulties experienced by early childhood educators in meeting the diverse needs of children. These difficulties include low educator-to-children ratios and insufficient knowledge or training to implement universal practices to support SEB+EF skills in early childhood classrooms. The report highlighted the need for greater awareness in the sector to enhance the provision of more equitable opportunities in early childhood care and education for children with a wide range of needs. This would include increasing early childhood educators' awareness about

children with disabilities and the specific needs of vulnerable or low-income families. Training is also required to build educators' skills and enhance their ways of working with these children and their families.

Early childhood programs that facilitate high-quality educator-child relationships and safe, stable learning environments are more likely to result in better social-emotional outcomes in children [11]. In an exploratory study in Singapore, early childhood educators ranked children's development of SEB skills as the most critical learning area in early childhood education. Yet, they also reported high professional development needs in their capacity to promote children's SEB skills and other pre-academic skills [12]. Considering findings from the Vital Voices for Vital Years 2 report [10] and the push to improve early childhood programming internationally [11], there is a need to expand efforts to improve early childhood educators' knowledge in developing children's SEB and EF as well as increasing educators' understanding of how to ensure their classroom practices are universal while catering for children with diverse needs.

Several published, validated, manualized school-wide interventions aimed at improving SEB+EF outcomes are used across a diverse range of countries (e.g., *Incredible Years* [13], *Second Step* [14]). However, importing and directly implementing interventions developed and tested overseas may not be acceptable or feasible for the early childhood sector in Singapore. Practices and concepts in early childhood interventions designed overseas must consider and incorporate Singapore educators' current practices and skillsets when implemented locally [9]. These interventions must be re-aligned to Singapore educators' unique needs, the local infrastructure, and existing learning frameworks. Without this, readiness to adopt new practices will likely be low [15], reducing the success of implementation and, in turn, the impact on children's outcomes.

## The EASEL Approach

To address the need to enhance early childhood educators' teaching practices and to ensure that these practices align with and fit with the local context, the Centre for Holistic Initiatives for Learning and Development (CHILD) has developed a universal classroom-based approach for early childhood educators, The Enhancing And Supporting Early development to better children's Lives (EASEL) Approach. The EASEL Approach consists of educator-led practices that can be incorporated into the daily classroom environment for children three to six years of age. This approach aims to enhance early childhood educators' teaching practices that promote children's SEB+EF development.

## Objectives

The EASEL trial has two objectives designed to accelerate the use of research evidence in practice. The first objective is to test the effectiveness of the EASEL Approach in enhancing educators' classroom practices to improve SEB+EF outcomes in preschool children. We hypothesize that the EASEL Approach will improve educators' teaching practices in promoting preschool children's SEB+EF outcomes. In turn, improvements in children's outcomes in these domains will also be observed at the end of the intervention (i.e., end of the school year). The research questions under this objective are: Does the EASEL Approach result in (1) improvements in educators' outcomes, namely teaching practices (e.g., quality of educator-child interactions, instructional support provided by educator, behavior management); and (2) improvements in children's SEB+EF outcomes as secondary outcomes?.

The second objective of the trial is to explore how educators implement the EASEL Approach within their classroom and identify factors that affect the implementation of the

EASEL Approach. We hypothesize that with the implementation strategies proposed for the trial (i.e., educator training, educator self-assessment of intention to use and self-efficacy, practice-based coaching, and data monitoring), the EASEL Approach will be adopted by educators, implemented sustainably, and achieve its desired effects. Research questions under this objective include: (1) how do educators use the EASEL Approach in the classroom, including the extent to which the EASEL Approach was integrated into educators' daily practice (i.e., penetration, fidelity); (2) how acceptable, feasible, and appropriate was the EASEL Approach for the early education context in Singapore; (3) what factors were associated with effective implementation and which implementation strategies were required for good uptake and adoption of the EASEL Approach; and (4) how sustainable is the EASEL Approach and how can it be scaled to other childcare sites nationwide? This would include identifying further adaptations or modifications required for the EASEL Approach to best meet the needs of early childhood classrooms and educators in Singapore.

## Methods and design

### Study design

The effectiveness of the EASEL Approach and its implementation will be evaluated with a type 2 hybrid effectiveness-implementation study, using a non-blinded, parallel groups, cluster randomized controlled design [16]. A type 2 hybrid study assesses both effectiveness outcomes (e.g., educator's teaching practices; children's SEB+EF outcomes) and implementation outcomes (e.g., fidelity; feasibility of the EASEL Approach), and is used when both the effectiveness of the intervention and the effectiveness of implementation are not well-known and need to be measured simultaneously. Specifically, four questions must be considered when selecting a hybrid study type [17]. First, what is known about the effectiveness of the intervention of interest? The EASEL Approach is rooted in the literature on effective classroom-based manualized interventions. These interventions, however, have not yet been implemented or tested in the Singapore context. Second, what is the extent to which an intervention needs to be adapted? In this case, there are substantial cultural and procedural differences in schools that could very well affect implementation and subsequent outcomes. EASEL's common elements approach to building a context-sensitive intervention allows for substantial adaptations to context while still delivering the essential components common to existing interventions. Third, what is the extent of knowledge about implementation determinants for the intervention in the context of interest? From the literature, we know that training alone is insufficient in transferring newly learned skills into applied use and that other strategies, such as coaching, will likely be required [18], thus suggesting a type 2 or 3 hybrid study. Finally, is the intervention ready for a "real world" application of implementation strategies. With the EASEL Approach, implementation strategies have been developed with relevant key partners, but we also need to focus on establishing the effectiveness of the intervention. Therefore, a type 2 hybrid study is recommended.

A cluster randomized controlled design allows for comparisons between childcare centers implementing the EASEL Approach with those continuing with business as usual. The study has been approved by the National University of Singapore (NUS) Institutional Review Board (NUS-IRB-2021-993).

### Procedure, setting, and participants

The sample for the EASEL trial will include 16 childcare centers in Singapore. Once recruited, the centers will be randomized to the intervention group (eight centers) or control group (eight centers) using a random number generator at a 50:50 allocation ratio. Due to the nature

of the intervention, there will be no concealment once the assignment has been made. Educators working at childcare centers in the intervention group will be trained and provided with ongoing coaching in the EASEL Approach. Those in the control group will continue with business as usual. Control group educators will be offered the opportunity to receive training in the EASEL Approach after the completion of the trial. After trial completion, primary caregivers of children in both intervention and control groups will receive a caregiver resource sheet briefly describing how they can implement the EASEL practices at home in interactions with their children.

We estimate that three classes in each childcare center will be eligible for the trial based on the age of the children (levels N2 –the second year of nursery where children are aged three years old at the start of the school year, K1 –the first year of kindergarten where children are aged four years old at the beginning of the school year, and K2 –second year of kindergarten where children are aged five years old at the start of the school year). In childcare centers with more than one class at each level, classes will be randomly selected for inclusion in the trial. We anticipate there will be approximately 15 children and one educator per class. The estimated target sample of children in the trial will be 600–700, and the estimated target sample of educators is 48 (one educator per class with at least one classroom per level in each childcare site) (see Fig 1).

Eligibility criteria are as follows: (1) the childcare center provides full-day programming for children at the N2, K1, and K2 levels (children aged three to five years old at the start of the

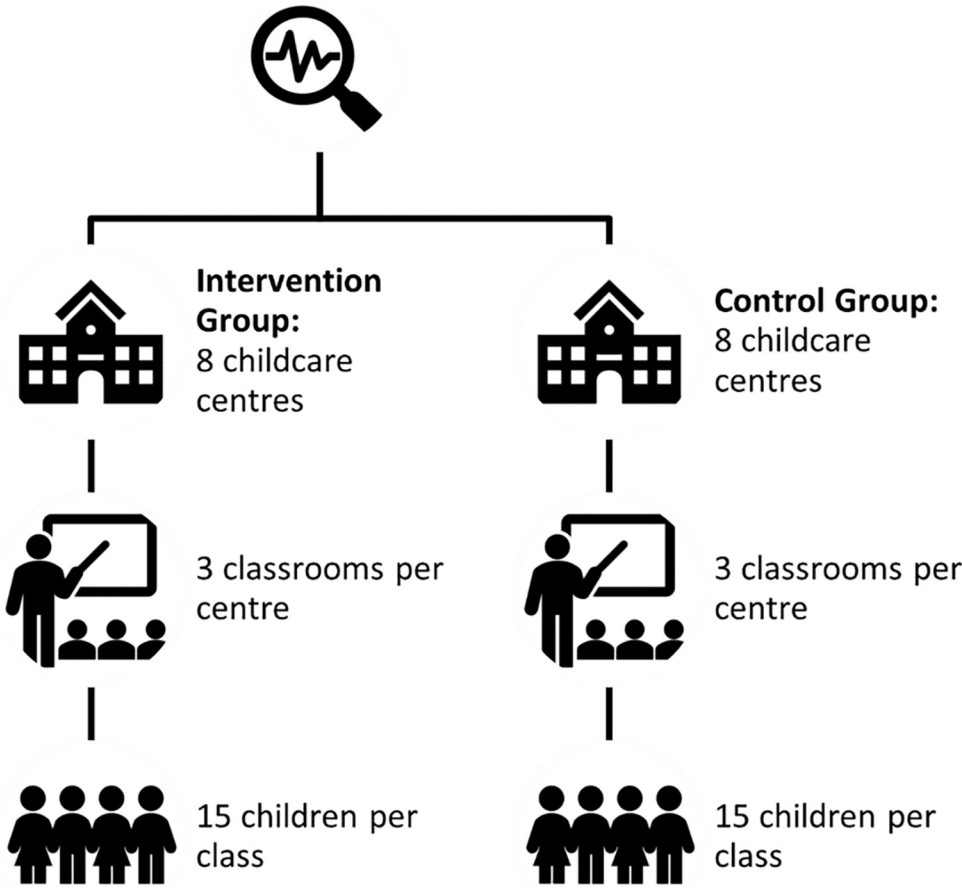

**Fig 1. Illustration of sample for EASEL trial.**

school year); (2) the childcare center has at least one of each class level that can participate in the EASEL trial; (3) the childcare center has close or equivalent to the national staff-to-child regulatory ratio of 1:20 for N2 classes, 1:25 for K1 classes and 1:30 for K2 classes; (4) the childcare center is not currently implementing other substantial interventions or involved in other trials evaluating either educator or student-focused interventions; (5) the educators have completed at least a diploma in early childhood care and education; and (6) the educators teach in English.

**Recruitment.** The research team will partner with local government agencies to identify and recruit early childhood center operators. In Singapore, large early childhood center operators can run and supervise multiple individual childcare centers. Once operators have expressed interest in participating in the EASEL trial, they will be asked to provide signed letters of commitment. Each operator will be asked to nominate several childcare centers participating in the trial (e.g., two centers for smaller operators and four centers for larger operators). Individual educators in nominated centers will be recruited–at least one educator per class level is required. The research team will conduct a short introductory workshop at each center for leadership staff and educators about the EASEL Approach and trial. All aspects of the research study will be explained in these sessions, and written informed consent will be obtained from individual educators at all sites, consistent with institutional review board protocols.

After the introductory workshop, the research team will provide childcare centers with an email or letter to be sent out to primary caregivers of children in classrooms participating in the trial. These documents will include an information sheet about the EASEL trial and a caregiver consent form. If required, follow-up emails and phone calls will be used to remind caregivers to complete their consent forms online if they wish to participate in the study.

## Development of the EASEL Approach

The educator-led practices in the EASEL Approach have been systematically drawn from effective classroom interventions using a common elements approach. The common elements approach offers an alternative way of synthesizing evidence for practice. It enables the design of interventions that can be easily tailored to the needs of different sites, practitioners, and children. Programs targeting the same outcome often consist of common core practices or techniques across programs. Common elements are these individual practices or techniques common across various programs that target a desired outcome and have demonstrated effectiveness [17]. Identifying common elements found in effective programs facilitates the promotion of a shared understanding of using evidence to improve SEB+EF outcomes in preschool children amongst early childhood educators. The common elements approach has been employed internationally in projects related to child protection, youth mental health, and early childhood [18–21].

In the EASEL Approach, common elements for use by educators were identified in the literature across manualized programs with evidence of effectiveness. These common elements or practices are used independently of the prescribed structure in many manualized programs, making them more accessible and flexible to implement in diverse settings.

The research team completed two systematic reviews to identify Common Elements from educator-led interventions to improve preschool children's SEB and EF outcomes. We built on McLeod et al.'s recent work [22] that identified educator-delivered practice elements to improve SEB outcomes of young children in early childhood classrooms. We updated this systematic review and conducted a separate systematic review using the same methodology for interventions targeting EF outcomes in preschool children to identify common elements for the EASEL Approach. These common elements were identified from programs that have

**Table 1. List of practices to be implemented in the EASEL trial.**

| EASEL practices | Outcomes targeted | | | |
|---|---|---|---|---|
| | S | E | B | EF |
| Active Listening | ✓ | ✓ | ✓ | ✓ |
| Rules & Routines | ✓ | ✓ | ✓ | ✓ |
| Constructive Feedback | | | ✓ | ✓ |
| Praise | ✓ | ✓ | ✓ | ✓ |
| Prompt, Wait Then Respond | ✓ | ✓ | ✓ | ✓ |
| Providing Choices | ✓ | ✓ | ✓ | ✓ |
| Encourage Child Ownership | | | | ✓ |
| Games Promoting EF (EF Playbook) | | | ✓ | ✓ |
| Dramatic Play | | | ✓ | ✓ |

S = social skills; E = emotional regulation; B = behaviors; EF = executive functioning

demonstrated effectiveness across multiple studies. A total of 25 common elements (referred to as EASEL practices) were identified across both reviews.

To increase the feasibility of implementing the EASEL practices (e.g., considering educators' time and resource constraints), a final list of nine EASEL practices was selected from the complete list of 25. The nine practices were chosen as core fundamental practices that are: (1) common across effective programs, (2) practices that expert stakeholders identified as ones for which local early childhood educators may need additional training, and (3) practices that are not already thoroughly covered in pre-service training to avoid replication for educators. Furthermore, the nine EASEL practices were compiled to ensure a balance of standard practices that local educators may already be implementing in their classrooms but can be enhanced and new practices for which educators require additional training before using. Finally, the nine EASEL practices were chosen to balance antecedent and consequential practices (i.e., practices that can be implemented before and after a target behavior) to ensure that educators both anticipate and respond to target behaviors.

The final list of nine EASEL practices was selected through an initial review by an internal team of experts with extensive experience in early childhood development and utilization of common elements approaches. A series of focus group discussions were also conducted with external experts in the early childhood education field (e.g., university lecturers of early childhood educators) to elicit their views on the terminology used in the EASEL Approach and practices that should be prioritized amongst local early childhood educators. The final set of practices to be implemented in the EASEL Approach is listed in Table 1, along with each practice's target SEB+EF outcome(s). The Executive Function Playbook (EF Playbook), developed by National University of Singapore, was used for one of the EASEL practices.

Practice Guides were written for each EASEL practice that identified the target behavior, suggested ways to use the EASEL practice, and provided examples of when to implement the practice in the classroom. The manual further includes information about how the EASEL Approach fits within Singapore's education frameworks: the NEL Framework and iTEACH principles. More details about the development process of the EASEL Approach are reported by Tan et al. [23].

## Use of the EPIS framework

The EASEL trial will be guided by the EPIS (Explore, Prepare, Implement, Sustain) Framework, an evidence-informed framework frequently used in program implementation and

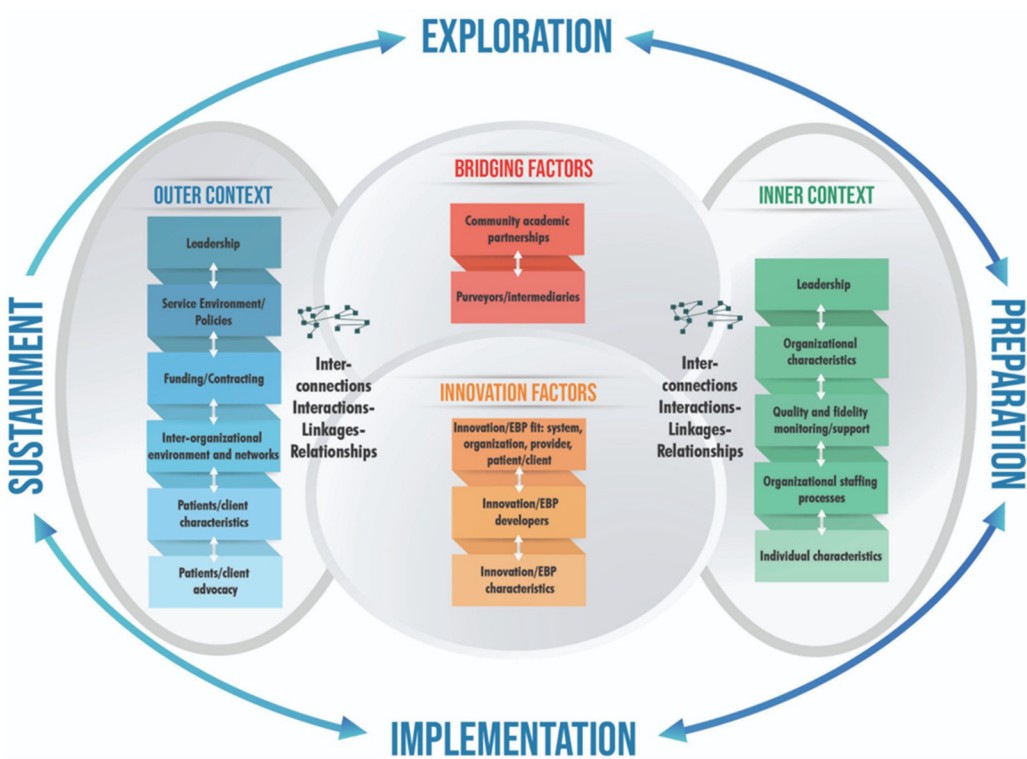

**Fig 2. EPIS framework** [45].

evaluation [24]. The EPIS Framework considers the multilevel nature of service systems (e.g., the early childhood sector in Singapore), the organizations within the service systems (e.g., individual early childhood center operators in Singapore), and the 'client' needs (e.g., the needs of children and early childhood educators) when implementing a new program. It highlights four key phases that guide and describe the implementation process: Exploration, Preparation, Implementation, Sustainment (see Fig 2).

Research has shown that attention to implementation factors before, during, and after the trial of a new practice leads to better implementation outcomes than if these factors were not deliberately considered [25, 26]. The EPIS Framework will guide the implementation and evaluation of the EASEL trial. See Fig 3 for the application of EPIS to the implementation and evaluation of the EASEL Approach. Implementation strategies and activities related to the evaluation trial are detailed under each of the four EPIS phases. Table 2 presents an implementation strategies matrix describing the implementation strategies for the EASEL Approach and the corresponding study period, EPIS stage, and parties involved.

## Phase 1: Exploration

The Exploration phase involves identifying the problem, developing appropriate and evidenced solutions, and considering factors that might impact implementation (e.g., acceptability, feasibility, and appropriateness of the EASEL Approach). In this context, the problem is the gap in educators' skills to enhance children's SEB+EF development, and the solution is the EASEL Approach.

**Stakeholder engagement.** Initial engagement meetings with community partners will be held to gather input about the acceptability and feasibility of the EASEL Approach. These

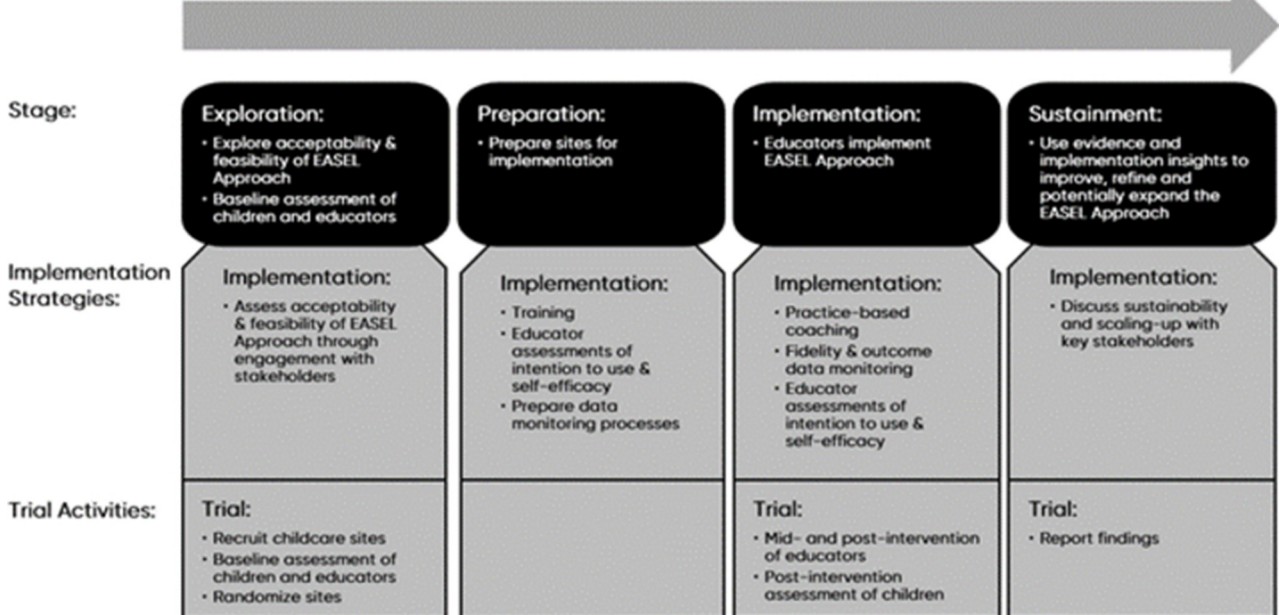

**Fig 3. Proposed flow of the rollout of the EASEL Approach and trial.**

consultations will typically involve key government representatives, early childhood center operators, childcare center leadership (e.g., principals), and educators. Regular meetings with key stakeholders will occur throughout all trial phases.

**Baseline assessment.** Baseline assessments for the trial will be conducted before randomization to ensure researchers are blind and naïve to the group condition. Outcome measures are detailed in the section on data collection.

## Phase 2: Preparation

The Preparation phase consists of a planning process with key stakeholders (e.g., relevant government agencies, early childhood center operators, and individual childcare centers) for implementing the new program by adapting and contextualizing the EASEL implementation plan with the Intervention Group childcare sites based on the information about feasibility gathered from stakeholders in the Exploration Phase.

**Processes for implementation and outcome data monitoring.** During this phase, the research team will establish processes for data monitoring with individual childcare sites. These processes will be incorporated into the local implementation plan for each childcare center. Data on the quality of implementation will be used in the ongoing coaching of

**Table 2. Implementation strategies matrix.**

| Implementation strategy applied | EPIS stage(s) |
| --- | --- |
| 1. Educator training | Preparation |
| 2. Educator self-assessment (e.g., intention to use, self-efficacy) | Preparation, Implementation |
| 3. Practice-based coaching using observational data and educator self-assessment | Implementation |
| 4. Implementation and outcome data monitoring for continual quality improvement | Implementation |

educators through the Implementation phase to stimulate reflection about barriers to implementation, and outcome data will be used to assess the effectiveness of the EASEL Approach.

**Training.**   All educators in the Intervention Group will receive training on the EASEL Approach. The training will be designed and delivered by the research team. This will involve a one-day in-person workshop for early childhood educators (format subject to negotiation with the participating childcare providers). Supplementary resources will also be available to educators online after the workshop for use throughout the trial and coaching sessions. These materials will be freely accessible through an online learning management platform, Thinkific.

## Phase 3: Implementation

When sufficient preparation has occurred, the Implementation phase commences (i.e., implementation of the EASEL Approach by educators in their classrooms). Factors affecting implementation include outer and inner contextual issues (e.g., resource availability, fit with educators' current practices, organizational culture, and attitudes) and consumer concerns (e.g., the applicability of practices for educators and children's needs).

**Implementation of the EASEL Approach.**   Early childhood educators will implement the EASEL Approach in their daily teaching practice to all children in their classrooms. Implementation support will be provided during ongoing coaching sessions.

**Practice-based coaching.**   Ongoing practice-based coaching of early childhood educators by CHILD coaches will occur throughout the EASEL trial. Although training can be helpful for teaching new skills or practices, training alone is insufficient in transferring newly learned skills into applied use [27]. Practice-based coaching bridges this gap between training and embedding the new skills into business-as-usual. Practice-based coaching occurs within the context of a collaborative partnership, with the cyclical approach generally involving: (1) shared goals and action planning, (2) focused observations, and (3) reflection and feedback [28]. Elek and Page [29] identified four critical features of effective coaching for early childhood educators through a comprehensive review of the literature. These features (i.e., observations, feedback, goal-setting, and reflection) will be incorporated into the practice-based coaching provided by the CHILD research team.

Coaching in the EASEL trial may take the form of regular group discussions with educators; a childcare center-based community of practice where educators at each site have a chance to discuss issues with the implementation of the EASEL Approach with facilitation from the research team; regular 'check-ins' between the educator and a coach; a review of online resources on Thinkific by the educator and a coach; and observation and feedback sessions where the coach conducts real-time observations and provides immediate feedback to the educator. The specific frequency and nature of the coaching in the EASEL trial will be determined after the training workshop through consultation with the childcare sites. The frequency of coaching will be at least fortnightly. The approach to coaching will also depend on the ability of the research team to enter childcare centers (based on prevailing COVID-related restrictions).

**Educator self-assessments.**   At regular intervals through the trial, educators will be asked to complete a short questionnaire to determine attitudes, perceived norms, intention to use the EASEL Approach, and self-efficacy about adopting the EASEL Approach. It is well-established that the strength of educators' intentions and attitudes about a new program or practice can influence the extent to which it is implemented [e.g., 30–33]. Educator self-assessments during the EASEL trial will allow the research team to understand potential barriers and facilitators to each educator's implementation of the EASEL Approach in their classrooms. The self-assessment questionnaire will be completed at three timepoints: upon completion of the

in-person training workshop, mid-intervention (6 months from the start of implementation), and post-intervention (at the end of the school year, i.e., approximately 9 months from the beginning of implementation). Measurement of self-assessment is described in more detail in the Measurement section.

**Implementation and outcome data monitoring.** The implementation of the EASEL Approach will be tracked through monitoring data. In addition to administrative data already being collected by childcare centers, this can include, but is not limited to, monitoring the frequency of use (uptake), adherence to the approach (e.g., through educator self-report and observational data by the research team) and the impact of the approach (e.g., outcome measures). Data monitoring will also include mid- and post-intervention assessments detailed in the Measurement section. Data monitoring allows the research team to understand the adoption and implementation of the EASEL Approach. Monitoring data can identify barriers and challenges to implementation as they arise rather than waiting for the trial to finish before addressing implementation issues [34].

## Phase 4: Sustainment

When the new program is routinely used, the Sustainment phase can begin. This involves the sustainment of the new program in the current setting and may also include scaling up to other settings and systems, using the evidence and implementation insights gained from the first three phases. Any further research or activity plans resulting from this work will be done in collaboration with key stakeholders. Findings from the trial will be shared with participating educators, and training on the EASEL approach will also be offered to educators in the Control Group.

## Data collection

A multilevel approach to measurement is necessary for our understanding of (1) the effectiveness of the EASEL Approach, and (2) the implementation of the EASEL Approach. Effectiveness outcomes of the EASEL Approach will be assessed at two data collection periods: baseline (i.e., at the start of the school year) and after implementation (i.e., at the end of the school year). Educator implementation outcomes will be assessed at three data collection periods: immediately post-training, midway through implementation, and post-implementation (i.e., at the end of the school year). See Table 3 for details of the measurements of implementation and effectiveness in the EASEL trial and Table 4 for more information on the assessments.

**Assessing effectiveness.** Most of the following measures have not been validated or used previously in Singapore but have often been evaluated elsewhere and were chosen due to their capacity for assessing key outcomes in Singapore as well as their requirements for administration (e.g., response burden, cost, training requirements).

*Early Childhood Classroom Observation Measure (ECCOM)*. Educators' teaching practices will be assessed on the ECCOM before training and post-intervention [35]. The ECCOM is an observational assessment that involves a 3-hour classroom observation by a trained researcher. Although the ECCOM has not previously been used or validated in Singapore, it has good face validity with locally used observational measures of classroom quality. Furthermore, the ECCOM is freely available and relatively easy to administer and code and thus can be factored into later sustainment and scaling-up plans if appropriate. See Table 5 for domains and subdomains of the ECCOM and EASEL practice(s) they map onto. The ECCOM subdomains of Literacy Instruction, Math Instruction, and Math Assessment will not be included because they are not the EASEL Approach's target outcomes.

**Table 3. Details of measurements and related outcomes in the EASEL trial.**

| Measure | Outcome | EPIS stage | Study period | | | | Key stakeholders involved |
|---|---|---|---|---|---|---|---|
| | | | Pre-training | Post-training | Mid-intervention | Post-intervention | |
| **Effectiveness** | | | | | | | |
| Early Childhood Classroom Observation Measure (ECCOM) | • Educators' teaching practices | Implementation | ✓ | | | ✓ | CHILD researcher |
| Child Self-Regulation & Social Behaviour Questionnaire (CSBQ) | • Child's social-emotional development<br>• Child's behavior<br>• Child's self-regulation skills | Implementation | ✓ | | | ✓ | Educators |
| Behaviour Rating Inventory of Executive Function–Preschool (BRIEF-P) | • Child's development<br>• Child's executive functioning | Implementation | ✓ | | | ✓ | Primary caregivers |
| The Singapore Whole Child Panel | • Child's pre-academic, theory of mind, executive functioning | Implementation | ✓ | | | | Children, CHILD researchers |
| **Implementation** | | | | | | | |
| Educator self-assessments | • Acceptability<br>• Appropriateness<br>• Uptake/adoption | Preparation, Implementation | | ✓ | ✓ | ✓ | Educators |
| Observational assessments by coaches (regularly) | • Uptake/adoption<br>• Fidelity | Implementation | | ✓ | ✓ | ✓ | CHILD coaches, educators |
| Student engagement data on Thinkific | • Uptake/adoption | Implementation | | | ✓ | ✓ | Educators |
| Focus Group Discussion | • Uptake/adoption<br>• Penetration<br>• Sustainability | Sustainment | | | | ✓ | Centre leadership, educators, key government representatives |

Researchers rate each domain on two dimensions: (1) child-centered practices and (2) educator-directed practices. Child-centered practices refer to educators' use of practices that provide shared responsibility with the child for both classroom management and learning, educators' active guidance, and support of children's learning efforts and their development. When educators implement child-centered practices, they provide clear, developmentally appropriate instructional goals that are balanced and integrated with the child's initiative and interests. A child-centered classroom environment is sensitive to and focused on children's needs and interests, but not to the extent that children have complete authority and control.

The second dimension of 'educator-directed practices' refers to practices that are controlled and directed to a large extent by the educator and focuses primarily on the acquisition of academic skills that the educator prioritizes. Rather than child-centered, an educator-directed

**Table 4. Time requirement for assessments in the EASEL trial.**

| Measure | Completed by | | | Duration |
|---|---|---|---|---|
| | Researcher | Educator | Primary Caregiver | |
| Educator self-assessment (of attitudes, perceived norms, intention to use, self-efficacy) | | ✓ | | 5 min per educator |
| Observational assessment by coaches of EASEL practices | ✓ | | | 3 hours per educator |
| Early Childhood Classroom Observation Measure (ECCOM) | ✓ | | | 3 hours per class |
| The Singapore Whole Child Panel (WCP) | ✓ | | | 20 mins per child |
| Child Self-Regulation & Social Behavior Questionnaire (CSBQ) | | ✓ | | 5–7 mins per child |
| Behaviour Rating Inventory of Executive Function–Preschool (BRIEF-P) | | | ✓ | 10–15 mins per child |

**Table 5. Mapping domains and subdomains of the ECCOM onto EASEL practices.**

| Domain | Subdomain | EASEL Practice(s) That Subdomain Maps Onto |
|---|---|---|
| Management | Child Responsibility | • Encourage Child Ownership<br>• Prompt, Wait Then Respond |
| | (Classroom) Management | • Rules & Routines |
| | Choices of Activities | • Providing Choices |
| | Discipline Strategies | • Constructive Feedback |
| Climate | Support for Communication Skills | • Active Listening<br>• Prompt, Wait Then Respond<br>• Dramatic Play |
| | Support for Interpersonal Skills | NA |
| | Student Engagement | • Providing Choices<br>• Active Listening |
| | Individualization of Learning Activities | • Praise |
| | Educator Warmth/Responsiveness | • Active Listening |
| Instruction | Learning Standards | • Encourage Child Ownership |
| | Coherence of Instructional Activities | • Games Promoting EF<br>• Dramatic Play |
| | Teaching Concepts | |
| | Instructional Conversation | • Active Listening<br>• Prompt, Wait then Respond |
| | Relevance of Activities to Children's Experience | NA |

Note. The scales for Literacy Instruction, Math Instruction and Math Assessment in the original ECCOM were excluded because these are not intended outcomes of the EASEL Approach.

curriculum is based on the educators' predetermined agenda. In educator-directed classrooms, peer interactions are minimized, and educators are more likely to impose solutions when conflicts arise rather than try to problem-solve with the children.

Researchers rate each domain on the two dimensions using a 5-point scale (1 = practices are rarely seen [<20% of the time], 2 = practices are not seen very much [20–40% of the time], 3 = practices are sometimes seen [40–60% of the time], 4 = practices are prominent [60–80% of the time], 5 = practices are predominant [80–100% of the time]). Domain scores will be averaged to generate a dimension score: Child-Centered Dimension Score and Educator-Directed Dimension Score.

Training on the ECCOM will involve a 4-hour training session to introduce researchers to the ECCOM dimensions and domains and general guidelines for observations and scoring. A second training session will include viewing two 30-minute videos of early childhood classrooms, and researchers will be asked to code them independently during the session. Ratings will be compared between coders, and discrepancies will be discussed. To be certified for reliability in coding the ECCOM, researchers will conduct a 3-hour practice coding with a senior coder in three early childhood classrooms that are not participating in the study. Ratings within one point of the senior coder's ratings reflect an acceptable degree of accuracy. During the data collection period, one in four early childhood classrooms will be double-coded by a senior coder with the same inter-rater reliability rules (i.e., a discrepancy of one point). If ratings are discrepant (i.e., more than one point difference), the researcher and senior coder will discuss to reach an agreement.

Child outcomes will be assessed through educator reports, caregiver reports, and assessments by the research team.

*Behavior Rating Inventory of Executive Functioning–Preschool version (BRIEF-P)*. To assess children's development and executive functioning, primary caregivers will be asked to complete the BRIEF-P at baseline (i.e., before implementation) and post-intervention (i.e., at the end of the school year) [36]. The BRIEF-P is a 63-item self-report questionnaire for primary caregivers to indicate how often a child has had problems with various behaviors in the past six months (0 = Never, 1 = Sometimes, 2 = Often). The BRIEF-P has been used to characterize children's EF in Singapore [37] and consists of the following indices: Inhibitory Self-Control Index, Flexibility Index, Emergent Metacognition Index, and Global Executive Composite; and the following domains: Inhibit, Shift, Emotional Control, Working Memory, Plan/Organize.

*Child Self-Regulation and Behavior Questionnaire (CSBQ)*. To assess children's social-emotional development, behavior, and self-regulation skills, educators will be asked to complete the CSBQ at baseline (i.e., before implementation) and post-intervention (i.e., at the end of the school year) [38]. The CSBQ is a 34-item self-report questionnaire for educators to indicate the option that best fits what each child is like (1 = Not True to 5 = Very True). Items on the CSBQ cover the following domains: Self-regulation (cognitive, emotional, behavioral), Sociability, Prosocial Behavior, Externalizing and Internalizing Behavior, and General Child Development.

*The Singapore Whole Child Panel (WCP) 2.2*. Children's EF will be screened at baseline using the EF tasks in the WCP 2.2 [39]. The WCP has been validated in the local population and developed with inputs from curriculum specialists in the education sector. This school readiness screening assessment will only be administered to children at baseline (i.e., before the implementation of the EASEL Approach). It will be used to discern which profiles of children improve more from the EASEL Approach (e.g., children with poorer or better EF at baseline).

The child will be brought to a quiet space in the childcare center, and the WCP 2.2 will be administered by a researcher using tablets. The assessment takes approximately 10 minutes, and the child will receive a small gift at the end of the administration.

**Assessing implementation.** *Educator self-assessments*. Early childhood educators will be asked to complete a short survey comprising four domains [30–33]: (1) their attitude toward the EASEL practices, (2) perceived norms of the EASEL practices, (3) intention to use the EASEL practices, and (4) self-efficacy with the EASEL practices. These four domains will be completed for each of the nine practices in the EASEL Approach. The self-assessments will be collected online by the research team. They will occur after completion of the training workshop (i.e., post-training), mid-intervention (i.e., 6 months), and post-intervention (i.e., end of the school year). Shorter versions focusing on specific EASEL practices may be completed regularly to inform reflections and learnings in coaching sessions.

*Observational assessments by coaches*. To facilitate reflection and feedback in coaching sessions, coaches will use a study-derived observational measure during regular classroom observations of early childhood educators' teaching practices. This measure will provide data on educators' fidelity to the nine EASEL practices (i.e., how often the educator implements the EASEL Approach in the classroom) and is rated on a 5-point scale (1 = practice is rarely seen, <20% of the time to 5 = practice is predominant, 80–100% of the time) for each practice. This measure is fully completed by coaches at three timepoints (i.e., baseline or first month of coaching, mid-intervention and post-intervention) and will be used as a measure of EASEL fidelity in the analyses.

*Student engagement data on Thinkific*. Data on the online learning management system, which houses the supplementary resource for the EASEL Approach, Thinkific, will be collected mid-intervention and post-intervention. These data allow us to assess the extent to which these online resources were accessed and completed by educators.

*Focus group discussion (FGD)*. At the end of the Implementation phase, FGDs will be conducted with groups of key stakeholders (e.g., center leadership and educators in the Intervention Group) to discuss the uptake/adoption during the trial, barriers to implementation, implementation strategies that helped prevent or resolve these barriers and sustainability (i.e., the usefulness of training and coaching for effective implementation of practices by educators) and scaling-up of the EASEL Approach. FGDs will be facilitated by members of the research team who were not involved in coaching educators.

## Data analysis

**Quantitative analysis.** Baseline data on effectiveness and implementation will be summarized along with demographic characteristics with descriptive statistics. The minimum, maximum, mean, and standard deviation will be reported for continuous variables. For ordinal and categorical variables, proportions will be reported. Between-group comparisons at baseline will be made to verify whether the randomization process was effective. Descriptive statistics for each subsequent timepoint will also be reported.

Intraclass correlations (ICCs) will be calculated for each outcome variable (i.e., educator teaching practices, child social-emotional outcomes) with baseline variables measured at the individual level by using childcare centre groupings as clustering variables to determine the proportion of the total variance attributable to specific childcare centre classrooms.

Separate ANCOVA multilevel models will be used to assess individual educator and child change scores between intervention and control groups while accounting for group membership (i.e., educators nested within childcare centers; children nested within both classrooms and childcare centers). Fixed or random effects models will be fit according to considerations of sample size, number of independent variables, and heterogeneity of sample. Heterogeneity of sample will be tested using either the Hausman or likelihood ratio test. We do not anticipate having to adjust for many individual differences as this is an RCT. However, we will test for baseline differences and, if such differences emerge, we will run sensitivity tests and adjust the analyses accordingly (i.e., statistically adjust for individual differences related to the outcomes).

Our primary outcomes will be educators' child-centeredness and educator-directedness on the ECCOM, and secondary outcomes will be children's SEB+EF outcomes on the BRIEF-P and CSBQ. EASEL fidelity from the classroom observational measure will be examined as a continuous mediator in the multilevel models. All outcome variables are continuous. A significance level of $p = 0.05$ will be used for all analyses.

*Sample size and power*. We drew on a meta-analysis by Luo and colleagues [40] to generate estimated effect sizes they estimated on child social competence, emotional competence, and challenging behaviors ($g$ = 0.42, 0.33, -0.31, respectively) [40]. Anticipated ICCs were drawn from findings on educators' teaching behaviors reported in Landry et al. [41] and Flook et al. [42]. Power analyses were conducted using Shiny CRT [43].

To achieve 80% power to detect a significant difference in educator teaching behaviors, a sample size of at least five clusters (i.e., childcare sites of three educators per site) in each condition is needed (i.e., a total of 10 clusters with 30 educators), assuming ICC is at 0.25. To achieve 80% power to detect a significant difference in child social-emotional outcomes, a sample size of at least two clusters (i.e., childcare sites of 45 children per site) in each condition is needed (i.e., a total of four clusters with 180 children), assuming ICC is at 0.22.

**Qualitative analysis.** A phenomenological approach will be used in the qualitative data analysis. Thematic analysis of data collected from coaching session notes and FGDs will be conducted using *a priori* codes. These codes will be based on domains of the EPIS framework,

with particular focus on the Inner Context and Innovation Factors, and the linkages between these factors. Thematic analysis will provide insight into potential barriers and facilitators to implementing the EASEL approach and the usefulness of the implementation strategies employed in the trial. Thematic analysis of the coaching notes will also provide insights about the influence of coaching on the implementation of the EASEL approach.

Two independent researchers will use the *a priori* codes to analyze the same FGD transcript. Coding will be discussed between the researchers to ensure consistency and to determine if additional codes need to be added to the codebook. One researcher will then analyze the remaining FGD transcripts and coaching notes. The second researcher will provide oversight and decision-making support. All qualitative analysis will be managed in qualitative analysis software (e.g., Dedoose). Using the coded data, both researchers will develop a thematic structure that focuses on the implementation of the EASEL Approach–barriers, facilitators, influence and effect of coaching, sustainment, and scalability.

**Mixed methods analysis.** The purpose of using mixed methods analysis in this study is that of complementarity–quantitative data will be used primarily to evaluate effectiveness while qualitative data will mainly be used to evaluate implementation of the EASEL Approach [44]. In general, quantitative and qualitative data will be collected in parallel (i.e., outcome measures and coaching notes), but the FGD with key stakeholders will be conducted at the conclusion of the trial. Data will be connected–that is, quantitative and qualitative data will be used to provide answers to related questions about the effectiveness and implementation of the EASEL Approach. Mixed methods analysis will be used to provide a more in-depth understanding of the barriers and facilitators to the implementation of the EASEL Approach. For example, quantitative outcomes using the ECCOM will be further explored using FGD data and educator self-assessments to provide additional insights about why there were or were not changes in the classroom environment after implementing the EASEL Approach. Child outcome data, reported by educators, will be connected to FGD data and coaching notes to provide some explanations for why children have or have not shown a change in their SEB+EF development.

## Data storage and management

The research team at CHILD, which is housed at the Yong Loo Lin School of Medicine, National University of Singapore, will oversee data management for this study with aid from the Data Management Team at the Singapore Institute for Clinical Sciences. Data will be stored on a database run from a locally hosted, secure server accessible only to CHILD researchers. As much as possible, data collection will be completed electronically and stored directly on the server. Where paper documents are used, data will be entered into the database after collection. The paper documents will be filed in a secure storage cabinet per guidelines from the Institutional Review Board. Focus group discussions will be recorded and transcribed. All audio recordings will be stored on the secure server and destroyed after data analysis.

All child and educator participants will be assigned a unique identifier. Identifiable information (e.g., name, contact details) will be stored separately and with highly secure and restricted access.

## Dissemination plan

Following the completion of the study, EASEL training materials and resources will be revised based on feedback from key stakeholders throughout the trial. The revised materials and

resources will be made freely accessible via an online platform to all early childhood educators in Singapore.

The research team will also take responsibility for the following activities in support of public dissemination: (1) register the awarded study with clinicaltrials.gov within 21 calendar days of the enrolment of the first participant, (2) submit the study's final results to clinicaltrials.gov within one year of the study's primary completion date, (3) disseminate the study results in peer-reviewed scientific journals, and abstract presentations at scientific conferences that are appropriate to the content of the study, (4) develop a community-centered fact sheet detailing the EASEL Approach and summarizing the key findings from the trial, and (5) present the study findings to local key stakeholders including but not limited to the Early Childhood Development Agency (ECDA), Ministry of Education (MOE) and National Institute of Early Childhood Development (NIEC).

## Discussion

The EASEL Approach will be evaluated using a cluster randomized controlled trial utilizing a type 2 hybrid effectiveness-implementation study. If the EASEL Approach is found to be successful, it can enhance Singapore's early childhood sector and equips educators to better support children in Singapore to gain the skills they need to succeed later in life. The first aim of the trial is to test the effectiveness of the EASEL Approach, a universal classroom-based approach, in enhancing educators' practices. The second aim of the trial is to understand how educators implement the EASEL Approach within their classrooms and the factors that affect implementation. At the end of this trial, we will have sufficient data to understand whether this approach is beneficial to the early childhood sector in Singapore and the barriers and facilitators for its implementation.

This study has significant potential benefits to the scientific community and communities of early childhood educators and preschool children. It is one of the first studies to use a common elements approach to develop an intervention for early childhood educators and the first to utilize a hybrid study to examine both the effectiveness and implementation of such an intervention. Findings from this study can inform the development of future interventions of this nature and enhance the professional development of early childhood educators, providing educators with the necessary skills and practices to promote children's development.

## Supporting information

**S1 Checklist. SPIRIT 2013 checklist: Recommended items to address in a clinical trial protocol and related documents\*.**
(DOC)

**S1 Fig. SPIRIT schedule of enrolment, interventions, and assessments.**
(DOC)

## Acknowledgments

We would like to acknowledge the significant contributions of the teams from the Early Childhood Development Agency (ECDA) for their advice and suggestions for the design for this trial.

Date and protocol version identifier: 27th July 2023 Version 1.1

Protocol amendments: Important amendments to the trial protocol will be communicated to investigators, NUS IRB, ClinicalTrails.gov registry and journal that the protocol is published in.

## Author Contributions

**Conceptualization:** Evelyn S. Tan, Bryce D. McLeod, Robyn A. Mildon, Cheryl K. F. Seah, Esther Goh, Gayatri Kembhavi.

**Methodology:** Evelyn S. Tan, Bryce D. McLeod, Robyn A. Mildon, Aron Shlonsky, Cheryl K. F. Seah, Keri McCrickerd, Esther Goh, Gayatri Kembhavi.

**Project administration:** Evelyn S. Tan.

**Supervision:** Bryce D. McLeod.

**Writing – original draft:** Evelyn S. Tan, Bryce D. McLeod, Gayatri Kembhavi.

**Writing – review & editing:** Evelyn S. Tan, Bryce D. McLeod, Robyn A. Mildon, Aron Shlonsky, Cheryl K. F. Seah, Keri McCrickerd, Esther Goh, Gayatri Kembhavi.

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
