## [Decision Letter · Decision Letter 0]

9 May 2023

PONE-D-23-00745Assessing the effectiveness and implementation of a universal classroom-based set of educator practices to improve preschool children's outcomes: Protocol for a cluster randomized controlled type 2 hybrid trial in SingaporePLOS ONE

Dear Dr. Tan,

Thank you for submitting your manuscript to PLOS ONE. After careful consideration, we feel that it has merit but does not fully meet PLOS ONE’s publication criteria as it currently stands. Therefore, we invite you to submit a revised version of the manuscript that addresses the points raised during the review process.

We look forward to receiving your revised manuscript.

Kind regards,

Qin Xiang Ng, MBBS, MPH

Academic Editor

PLOS ONE

Journal Requirements:

Reviewers' comments:

Reviewer's Responses to Questions

**Comments to the Author**

1. Does the manuscript provide a valid rationale for the proposed study, with clearly identified and justified research questions?

Reviewer #1: Yes

Reviewer #2: Partly

Reviewer #3: Partly

2. Is the protocol technically sound and planned in a manner that will lead to a meaningful outcome and allow testing the stated hypotheses?

Reviewer #1: Partly

Reviewer #2: Partly

Reviewer #3: Yes

3. Is the methodology feasible and described in sufficient detail to allow the work to be replicable?

Reviewer #1: Yes

Reviewer #2: Yes

Reviewer #3: Yes

4. Have the authors described where all data underlying the findings will be made available when the study is complete?

Reviewer #1: No

Reviewer #2: Yes

Reviewer #3: Yes

5. Is the manuscript presented in an intelligible fashion and written in standard English?

Reviewer #1: Yes

Reviewer #2: Yes

Reviewer #3: Yes

6. Review Comments to the Author

You may also provide optional suggestions and comments to authors that they might find helpful in planning their study.

Reviewer #1: This article presented a protocol for a cluster randomized trials to assess the effectiveness and implementation of a universal classroom-based set of educator practices to improve preschool children's outcomes in Singapore. This study has followed the protocol checklist for clinical trials. However, I have several comments and hope the authors can clarify them.

1. This protocol has no clearly defined primary outcomes. I understand that the authors have mentioned in the manuscript that “Primary outcomes include educators' teaching practices and their adoption of the EASEL Approach in everyday practice”. Under the “Assessing effectiveness” section, there are several outcomes, which are potentially the outcomes to assess the EASEL approach in everyday practice. Are them all co-primary outcomes?

2. Following the first question, it is unclear how the sample size calculation was done. Which outcome it was based on? “25% difference in child outcomes” refers to which outcome? Since it is a cluster randomized trial, how the clustering effects (e.g., what were the assumed ICCs or variance components) were considered in the sample size calculations? I notice that there are several places (pages 8 and 23) were mentioning the sample sizes. I would like to suggest to have a “sample size and power” section to (at least) include the following information

a. Outcome used for power calculation

b. Assumed effect size

c. Cluster size

d. Number of clusters

e. Assumed ICCs (or other measure of clustering effect)

3. A more detailed description of statistical analysis is needed. There seems to be many outcomes measured. Will all of them be analyzed using MLMs. What kind of MLM models the authors is planning to use. Will the authors use the change scores as the outcome, or an ANCOVA-like approach will be used? Will the analyses be unadjusted (since there is no covariates-adjustment mentioned in the section)?

4. If there are multiple co-primary outcomes, adjustment for multiplicity is needed

5. There are many places with “Error! Reference source not found”. Please check the links inserted in the manuscript.

Reviewer #2: I read this study protocol with interest but the study outcomes are rather ill-defined at the moment. The overall design may be too ambitious as well for a single study. Please see below for my specific comments.

Specific comments:

1. The study title requires further specification. What outcomes exactly? Are we examining social emotional competence and learning outcomes? Please be specific.

2. “In an exploratory study, early childhood educators in Singapore ranked children's development of SEB skills as the most critical learning area in early childhood education. Yet, they also reported high professional development needs in their capacity to promote children's SEB skills along with other pre-academic skills (12). Taken together with findings from the Vital Voices for Vital Years 2 report (11), there is a need to expand efforts to improve early childhood educators' knowledge in developing children's SEB and EF as well as increasing educators' understanding of how to ensure their classroom practices are inclusive of children with diverse needs” - much of the information presented in the introduction is highly localized, suggest rephrasing and making it more succinct to make it reader friendly for a wider audience

3. Although the authors gave some reasons to support the choice of the EPIS (Explore, Prepare, Implement, Sustain) Framework, it is unclear why this framework was chosen over others, e.g. the dynamic sustainability framework. In my reading of the manuscript, this seems like a viable choice given that interventions are not static and can be continually improved. What is the main outcome of the study? Suggest making a stronger link between the framework of choice and the intended primary outcome.

4. Several instances of “Error! Reference source not found” throughout the manuscript. Sloppy.

5. “Multilevel Models (MLMs) will be used to assess differential patterns of change over time in each effectiveness measure between intervention and control groups across baseline and later timepoints while accounting for and measuring the effect of nonindependence of individual subjects (i.e., educators nested within childcare centers; children nested within both classrooms and childcare centers)” - this is very broad and generic statement. What MLM models exactly? Are you dealing with ‘aggregated’ or ‘collapsed’ variables? Scaled weight analyses or unweighted analyses? Further elaboration and details are missing here. It is prudent to state and adjust for foreseeable confounders in the final analysis.

6. “Data will be stored on a bespoke database run from a locally-hosted, secure server that is accessible only to CHILD researchers” - what exactly is this? Is this something created specifically for the purposes of the study?

7. Would the authors have close monitoring of attrition and adjust the sample size accordingly?

Reviewer #3: In the manuscript ‘*Assessing the effectiveness and implementation of a universal classroom-based set of educator practices to improve preschool children's outcomes: Protocol for a cluster randomized controlled type 2 hybrid trial in Singapore*’, the authors describe the development of a new intervention (EASEL Approach) for 3-6 yr old children in Singapore and the design of an evaluation which focuses on both implementation and effect at teacher and child level (type 2).

p. 3: The contribution of this study to the literature is not yet fully clear. Only the last bullet point seems directly related to ’contribution to the literature’. The authors should indicate here: what will be the new knowledge after completion of this study?

Abstract, Introduction, Discussion: I doubt whether an ECEC program can ‘equip’ 3-6 year old children with competences and skills. ECEC programs can stimulate children’s development and skills at an early age and these skills are significant precursors of more holistic, broader skills at elementary school. I only suggest to use vocabulary that is developmentally appropriate, i.e. fits in with insights from developmental psychology and educational science.

p. 4: What does ‘inclusive’ mean in the context of this manuscript? The word ‘inclusion’ has been used in various ECEC contexts (e.g., immigrant families, children with autism, challenging but normal behaviour of toddlers, etc.). The manuscript suggests that certain children are challenging for ECEC teachers in Singapore. However, in the design, sample, or analysis, there is no explicit link to certain groups of children (see also below).

Table 1: The authors describe the selection process for the 9 EASEL themes with detail (see Table 1). Could they also report the scores for criteria 1, 2 and 3? Which instructional behaviors are still considered challenging for many ECEC teachers in Singapore?

p. 6: This question may be somewhat difficult, but I would like to know the answer. There is a new intervention that is not grounded with a specific theory of framework, but that seems to be designed in a systematic but also eclective approach. What do the 9 selected themes from the EASEL intervention represent? Is it a bottom-up selection by an expert committee and/or is it a selection of themes that are generally inadequate in regular ECEC in Singapore? Is it a coherent set or a heterogeneous blend of themes, is it a list of 9 separate items or is there perhaps a further categorization possible? For example, are the themes mostly related to instructional support or is it related to instructional support, classroom organization and/or emotional support? Or using a different categorization, which themes are linked with SEB and which ones with EF (see p. 3 for these abbreviations). And can the set of 9 themes be trained in a similar instructional format?

p. 7: using A non-blinded à a (i.e., without capital)

p. 9, Design & sample: The authors do not mention attrition at center, teacher or child level. Is this realistic?

p. 13: preparation has occurred à I am not a native speaker, but is this grammatically correct?

p. 14: naïve à no “ï”?

p. 15-19: In this part, there seems to be some redundancy. The implementation theme is repeated in this part of the text. Could the authors write this part in a more concise way / with less repetition? 

p. 17: What are the current Covid19-restrictions in Singapore ECEC centers?

p. 20: The authors have selected the ECCOM measure (Early Childhood Classroom Observation Measure), but readers cannot accurately judge from this design paper whether / why this is a good choice. This is, first of all, relate to my abovementioned comment related to the exact content of the EASEL intervention, which was not fully clear to me. Second, the authors indicate which subscales from the ECCOM are not selected, but the reader cannot find which subscales from the ECCOM are considered relevant in this evaluation of EASEL.

Are the ECCOM raters (i.e., the CHILD researchers from the Centre for Holistic Initiatives for Learning and Development) blind to experimental condition? Or are they relatively close to the research team and/or the EASEL intervention so that they are aware and may recognize the location and intervention?

If random assignment is “fluked” in this relatively small sample (i.e., at center level), what will be the strategy of the researchers? Will they include covariates? Will they apply other statistical analyses to control for possible confounding?

p. 23: Could the authors give an indication of the design effect (ICC) for teachers and children? How does this affect the statistical power of their analysis?

p. 23: Could the authors qualify the a priori assumed difference of 25%: is this a small, medium or large effect according to the rules of thumb of Cohen (1988)? The authors could actually base their sample on effect sizes from previous experimental studies into this domain. The challenge is that meta-analyses have demonstrated small-to-medium differences at process quality/teacher/center level, whereas the experimental effects for children seem generally smaller (see, for example, Egert et al., 2018). I can imagine that statistical power is not the same for the research questions at teacher level (a) and at child level (b).

p. 23: The proposed multi-level analysis takes into account that educators are nested within childcare centers. However, the number of centers and, relatedly, the number of teachers seems relatively small. Perhaps the authors could consider to include teachers as a factor which may be explored in a preliminary analysis (instead of a level).

p. 23: See my previous comment: The authors introduce the importance of challenging behaviors of some students in an inclusive ECEC setting. I had expected therefore that the analysis would include a child factor that would capture this child characteristic instead of a universal, but perhaps too general effect for all children.

p. 23: The authors emphasize the importance of implementation. Do I understand correctly that implementation is not a moderator in their analysis? If so, I suggest that a continuous moderator seems vital for the analysis of a newly developed and implemented intervention? I emphasize the continuous nature of such a variable, because I read that ‘why there were of were not changes’ as an all-or-nothing matter.

p. 24: Qualitative - or: Qualitative analysis

p. 24: The thematic content analysis is not described with adequate detail. The authors could describe the qualitative analysis with the general methodological approach, *a priori* themes in a coding scheme, use of software, description of coding and training coders, qualitative analysis after coding (etc.).

p. 24: The triangulation and mixed methods analysis is not described with adequate detail. Triangulation suggests that two different types of data (i.e., qualitative & quantitative) are related to a similar construct (i.e., implementation of EASEL). My question is: is there a order in the analysis? Which analysis may follow and build upon the other? And my other question is: what will the researchers do when the different types of data do not converge? Terminology like *convergent / parallel / embedded / sequential / exploratory vs. explanatory* may give a more accurate picture of the planned analysis.

By the way, there were some missing links missing (“Error!”) in the manuscript (e.g., p. 12 and 14).

A final note:

I take the liberty to share some of my thoughts related to the intervention and its implementation. The authors indicate that a kick-off with the research team and a one-day workshop is the starting point of the intervention in practice. Coaching will be the next phase and the authors indicate that they want to identify barriers and challenges with the implementation based on meetings with ECEC teachers.

My thoughts / concerns are as follows:

There is no (small-scale) pilot. Hence, learning about barriers and challenges takes place during the evaluation with measures at teacher and child level.A one-day workshop seems very brief, even it is one-on-one.The coaching seems very helpful and, in my opinion, is seriously needed for effective implementation at teacher level. The authors refer to Elek and Page (2019), which is an important source. Also other reviews have (recently) been published related to coaching in an ECEC context (e.g., Blewitt et al., 2011; Donath et al., 2023; Egert et al., 2018; Obee et al., 2022; McLeod et al., 2023; Young et al., 2021). These new sources are interesting for the readers of this paper (a) and they may provide more guidance in the design and delivery of the coaching part of the newly developed intervention (b).I hope that the research team will have constructive dialogues with the teachers as experts in their own classroom and with their students.

I have added these comments from my background in ECEC pre-service and in-service training and other forms of professional development, including coaching, and I would like to understand this new intervention a little bit better. The question behind this question (i.e., my concern) is that it may be very difficult for the research team, the instructor in the one-day session, the coach in the period thereafter, and, most of all, for the ECEC teacher – after a relatively brief training – to implement various changes in his/her behavior throughout the day in relatively large classrooms with young children. To be perfectly clear, my questions are linked to positioning the intervention for readers of the journal, and they may also be interpreted as motivations for this study.

Finally, I look forward to read about the findings of this experimental study.

7. PLOS authors have the option to publish the peer review history of their article (what does this mean?). If published, this will include your full peer review and any attached files.

Reviewer #1: No

Reviewer #2: No

Reviewer #3: No

---

## [Author Response · Author response to Decision Letter 0]

27 Jul 2023

Dear Editor and Reviewers, 

We thank you for your detailed and constructive comments to help us improve on our manuscript. We have responded to all of your comments in the Response to Reviewers document. Please let us know if you have any further questions with regard to our responses and our manuscript. 

Many thanks,

Evelyn Tan on behalf of all co-authors

---

## [Decision Letter · Decision Letter 1]

22 Aug 2023

PONE-D-23-00745R1Assessing the effectiveness and implementation of a universal classroom-based set of educator practices to improve preschool children's social-emotional outcomes: Protocol for a cluster randomized controlled type 2 hybrid trial in SingaporePLOS ONE

Dear Dr. Tan,

Thank you for submitting your manuscript to PLOS ONE. After careful consideration, we feel that it has merit but does not fully meet PLOS ONE’s publication criteria as it currently stands. Therefore, we invite you to submit a revised version of the manuscript that addresses the points raised during the review process.

We look forward to receiving your revised manuscript.

Kind regards,

Qin Xiang Ng, MBBS, GDMH, MPH

Academic Editor

PLOS ONE

Journal Requirements:

Reviewers' comments:

Reviewer's Responses to Questions

**Comments to the Author**

1. Does the manuscript provide a valid rationale for the proposed study, with clearly identified and justified research questions?

Reviewer #1: Yes

2. Is the protocol technically sound and planned in a manner that will lead to a meaningful outcome and allow testing the stated hypotheses?

Reviewer #1: Yes

3. Is the methodology feasible and described in sufficient detail to allow the work to be replicable?

Reviewer #1: Yes

4. Have the authors described where all data underlying the findings will be made available when the study is complete?

Reviewer #1: Yes

5. Is the manuscript presented in an intelligible fashion and written in standard English?

Reviewer #1: Yes

6. Review Comments to the Author

You may also provide optional suggestions and comments to authors that they might find helpful in planning their study.

Reviewer #1: I thank the authors to address my comments. I believe most of my comments have been well-addressed. There is one minor issue that I would like the authors to clarify.

on page 24-25, the authors mentioned "Random intercepts and slopes by subject (e.g., child or educator) will be tested and used, if necessary. Otherwise, we will use fixed effects to conserve statistical power." It seems to be a bit vague to me. Can the authors be more specific on this? What does it mean "if necessary" and when the fixed effect model will be applied.

7. PLOS authors have the option to publish the peer review history of their article (what does this mean?). If published, this will include your full peer review and any attached files.

Reviewer #1: No

---

## [Author Response · Author response to Decision Letter 1]

30 Aug 2023

Dear Editor and Reviewer, 

Thank you for your comments. We have responded to them in the 'Response to Reviewers' document and submitted the manuscript with tracked changes and the final manuscript. 

We have also addressed the 4 issues detailed in the email dated 31st August 2023. Many thanks. 

Kind regards,

Evelyn Tan and on behalf of my co-authors

---

## [Editor Report · Decision Letter 2]

6 Sep 2023

Assessing the effectiveness and implementation of a universal classroom-based set of educator practices to improve preschool children's social-emotional outcomes: Protocol for a cluster randomized controlled type 2 hybrid trial in Singapore

PONE-D-23-00745R2

Dear Dr. Tan,

We’re pleased to inform you that your manuscript has been judged scientifically suitable for publication and will be formally accepted for publication once it meets all outstanding technical requirements.

Kind regards,

Qin Xiang Ng, MBBS, GDMH, MPH

Academic Editor

PLOS ONE
---

## [Editor Report · Acceptance letter]

11 Sep 2023

PONE-D-23-00745R2 

Assessing the effectiveness and implementation of a universal classroom-based set of educator practices to improve preschool children's social-emotional outcomes: Protocol for a cluster randomized controlled type 2 hybrid trial in Singapore 

Dear Dr. Tan:

I'm pleased to inform you that your manuscript has been deemed suitable for publication in PLOS ONE. Congratulations! Your manuscript is now with our production department. 

Kind regards, 

on behalf of

Dr. Qin Xiang Ng 

Academic Editor

PLOS ONE